# Peptidoglycan reshaping by a noncanonical peptidase for helical cell shape in *Campylobacter jejuni*

Kyungjin Min [1,7], Doo Ri An[2,3,7], Hye-Jin Yoon[1,7], Neha Rana[4,7], Ji Su Park[1], Jinshil Kim[5], Mijoon Lee [4], Dusan Hesek[4], Sangryeol Ryu [5,6], B. Moon Kim[1], Shahriar Mobashery[4]*, Se Won Suh[1,2]* & Hyung Ho Lee[1]*

Assembly of the peptidoglycan is crucial in maintaining viability of bacteria and in defining bacterial cell shapes, both of which are important for existence in the ecological niche that the organism occupies. Here, eight crystal structures for a member of the cell-shape-determining class of *Campylobacter jejuni*, the peptidoglycan peptidase 3 (Pgp3), are reported. Characterization of the turnover chemistry of Pgp3 reveals cell wall D,D-endopeptidase and D,D-carboxypeptidase activities. Catalysis is accompanied by large conformational changes upon peptidoglycan binding, whereby a loop regulates access to the active site. Furthermore, prior hydrolysis of the crosslinked peptide stem from the saccharide backbone of the peptidoglycan on one side is a pre-requisite for its recognition and turnover by Pgp3. These analyses reveal the noncanonical nature of the transformations at the core of the events that define the morphological shape for *C. jejuni* as an intestinal pathogen.

[1] Department of Chemistry, College of Natural Sciences, Seoul National University, Seoul 08826, Korea. [2] Department of Biophysics and Chemical Biology, College of Natural Sciences, Seoul National University, Seoul 08826, Korea. [3] Hazardous Substances Analysis Division, Gyeongin Regional Office of Food and Drug Safety, Incheon 22133, Korea. [4] Department of Chemistry and Biochemistry, University of Notre Dame, Notre Dame, Indiana, IN 46556, USA. [5] Department of Food and Animal Biotechnology, Department of Agricultural Biotechnology, and Research Institute for Agriculture and Life Sciences, Seoul National University, Seoul 08826, Korea. [6] Center for Food and Bioconvergence, Seoul National University, Seoul 08826, Korea. [7] These authors contributed equally: Kyungjin Min, Doo Ri An, Hye-Jin Yoon, Neha Rana. *email: mobashery@nd.edu; sewonsuh@snu.ac.kr; hyungholee@snu.ac.kr

The human pathogens *Campylobacter jejuni* and *Helicobacter pylori* are well-studied members of Epsilon proteobacteria, which almost exclusively is comprised of species with a curved/helical morphology[1]. Although some rod-shaped *Campylobacter* species have been described and rod-shaped variants of *C. jejuni* have been isolated, the curved/helical shape is the standard morphology of *C. jejuni*[1]. The helical shape of *C. jejuni* is important for bacterial colonization during infection, to move through the mucus layer of the gastrointestinal tract, and for entry into host cells by a corkscrew-like motility[2]. Motility of *C. jejuni* is a critical factor for host colonization and pathogenesis, with non-motile strains being severely impaired in their ability to colonize the host intestines. Thus, inhibition of proteins responsible for the helical cell shape could be useful in interference with the bacterial lifestyle and virulence[2–4]. *C. jejuni* infection is considered to be the most prevalent cause of bacterial diarrheal diseases worldwide, triggering severe complications such as inflammatory bowel disease, reactive arthritis, and Guillain-Barré syndrome[5,6].

The helical shape of *C. jejuni* is believed to be due to the specific type of crosslinking of peptidoglycan[7,8]. Bacterial peptidoglycan is mainly comprised of a linear polysaccharide chain consisting of repeating β-(1→4)-linked *N*-acetylglucosamine (GlcNAc or NAG)-*N*-acetylmuramic acid (MurNAc or NAM) disaccharide unit, with a pentapeptide attached to the NAM[9]. In *C. jejuni*, the pentapeptide sequence is L-Ala$^1$-γ-D-Glu$^2$-*m*DAP$^3$-D-Ala$^4$-D-Ala$^5$, where *m*DAP refers to *meso*-2,6-diaminopimelate. The neighboring peptidoglycan strands are further crosslinked exclusively by the 4→3 amide linkage between the main chain of D-Ala$^4$ from one strand and the side chain of *m*DAP$^3$ from another strand[10–12] (Fig. 1a). There are a host of enzymes that modify the peptidoglycan[13,14]. In the case of *C. jejuni*, cell-shape-determining (Csd) proteins are critical[9]. To elucidate how the peptidoglycan is remodeled by these enzymes, their structural and functional characterizations are important. Crystal structures of the Csd3 monomer and Csd2 homodimer, as well as a heterodimer between Csd1 and Csd2 from *H. pylori*, have been determined, and structural comparisons have revealed that the three lysostaphin-like metalloprotease (LytM) domains show high similarity in the overall structures[15,16]. Notwithstanding the earlier studies, the mechanism by which Csd proteins recognize their peptidoglycan substrate is unknown. Thus, it is necessary to determine the structures of Csd proteins in complex with their substrates.

Most *C. jejuni* strains have several LytM homologs including orthologs to *H. pylori* Csd gene products. Two LytM homologs, Pgp1 (peptidoglycan peptidase 1) and Pgp2 (peptidoglycan peptidase 2) from *C. jejuni*, were identified as being orthologous to Csd4 and Csd6 from *H. pylori*, respectively[17,18]. Pgp1 and Pgp2 from *C. jejuni* are also responsible for cleaving peptidoglycan stem peptides by their carboxypeptidase activities[19,20]. Pgp1 is a D,L-carboxypeptidase and cleaves tripeptide stems into dipeptides, whereas Pgp2 cleaves tetrapeptide stems into tripeptides as an L,D-carboxypeptidase[19,20]. Mutations in the genes of Pgp1 or Pgp2 alter the muropeptide profile leading to inability to form the helical cell shape[19,20].

In the present report, we have characterized structurally and functionally the *A8118_01115* gene product (274 amino acids) from *C. jejuni*, and assigned it to the LytM protease family. We disclose that the *A8118_01115* gene product exhibits both the D,D-endopeptidase and D,D-carboxypeptidase activities. The *A8118_01115* gene product is distinct structurally from both Pgp1 and Pgp2, which we hereby designate as peptidoglycan peptidase 3 (Pgp3). We report eight X-ray crystal structures of Pgp3, which show the protein in two unique conformations with an open and a closed active site. The structures are: (i) tartrate-bound wild type (WT) (open form), (ii) citrate-bound WT (two structures, closed form), (iii) H216A mutant (open form), (iv) H216A mutant with tartrate (closed form), (v) H247A mutant (open form), (vi) H247A mutant with pentapeptide 1, Ac-L-Ala$^1$-γ-D-Glu$^2$-*m*DAP$^3$-D-Ala$^4$-D-Ala$^5$ (open form) (Supplementary Fig. 1), and (vii) H247A mutant with crosslinked muramyl tetra-tri peptide 2, NAM-L-Ala$^1$-γ-D-Glu$^2$-*m*DAP$^3$-D-Ala$^4$-*m*DAP$^3$-γ-D-Glu$^2$-Ala$^1$-NAM (open form) (Supplementary Fig. 2). Tetra-tri peptide in 2 is a minimalist D,D-crosslinked substrate. Despite our efforts to crystallize the WT enzyme with peptidoglycan substrates, the crystals turned over the substrates, which made the task impossible. The structures of the complexes were determined with an inactive mutant variant. This report describes the enzymatic activity of Pgp3, its conformational states, domain architecture, and comparison with another M23 metallopeptidase (i.e., LasA virulence factor from *Pseudomonas aeruginosa*; PDBs 3IT5 and 3IT7), in addition to a catalytically inactive variant of Pgp3 in complex with peptidoglycan substrates. The analysis provides insight into the catalytic mechanism of this important bacterial enzyme.

## Results

**Pgp3 has D,D-carboxy- and D,D-endopeptidase activities**. In order to investigate the role of Pgp3 in peptidoglycan turnover, we performed assays using seven synthetic muramyl peptide substrates (compounds 2–8), each prepared in multistep syntheses by known procedures[21–23] (Supplementary Fig. 2 and Supplementary Methods). We observed that Pgp3 exhibited both D,D-carboxypeptidase and D,D-endopeptidase activities based on LC/MS analysis (Fig. 1b). We were not able to isolate the sacculus of *C. jejuni* in quantity. However, the peptidase activity was also documented with the *Escherichia coli* sacculus as its macromolecular substrate (Supplementary Fig. 3). The mature *E. coli* sacculus is devoid of the full-length pentapeptide, so the D,D-carboxypeptide activity could not be monitored, but we documented the D,D-endopeptidase activity (Supplementary Fig. 3). Increasing the concentration of Pgp3 diminishes the D,D-crosslinked peptide (colored in blue, green, and red in Supplementary Fig. 3).

**The *pgp3* deletion affects helical morphology of *C. jejuni***. To study the role of *pgp3* in *C. jejuni* morphology, we constructed a deletion mutant and the morphology of the *pgp3*-deletion mutant was examined by transmission-electron microscopy (TEM). The *pgp3*-deletion mutant exhibits a curved-rod appearance, a significant deviation from the helical morphology of the WT (Fig. 1c). When complementation was attempted by expressing the *pgp3* gene, the characteristic *C. jejuni* helical shape was restored (Fig. 1c). We next investigated the role of *pgp3* in an invasion assay using Caco-2 cells, as a measure of virulence. The *pgp3*-deletion mutant exhibited significantly decreased invasive ability compared to the WT (Fig. 1d), while the invasive ability was restored appreciably after complementation (Fig. 1d). These results suggest that Pgp3 is important for maintenance of *C. jejuni* helical shape and to its virulence.

**Crystal structure of Pgp3 with open and closed conformations**. In order to gain mechanistic insight into how Pgp3 works, we have determined the X-ray structures of the WT Pgp3 in two crystal forms (bound separately to tartrate and to citrate). In both structures, the crystallographic asymmetric unit contained two Pgp3 monomers (chain A and chain B) with space groups of $P6_1$ and $P3_221$, respectively (Supplementary Table 1). For the tartrate-bound WT, a model accounting for all the amino acids of the recombinant Pgp3 (Glu21-Asn273) was refined at a resolution of

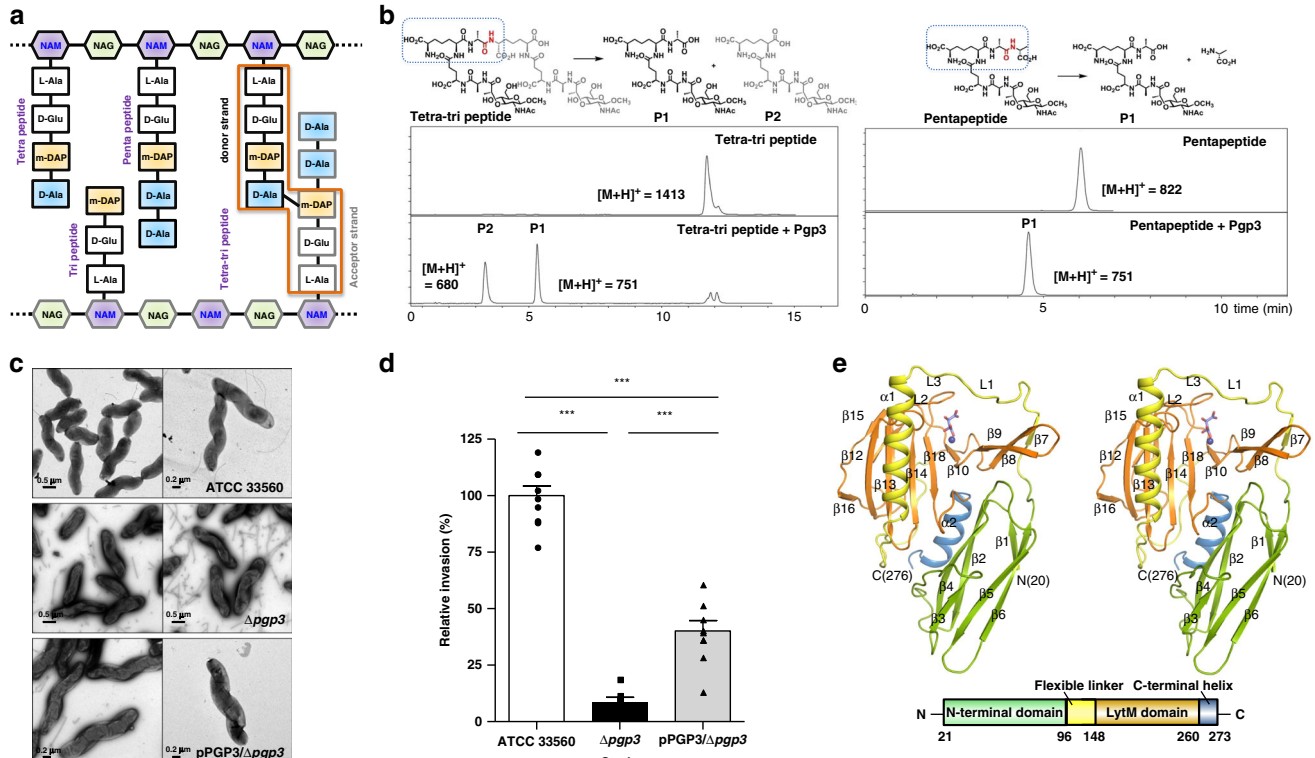

**Fig. 1 Structural and functional characterizations of Pgp3. a** Schematic diagram of peptidoglycan structure. **b** LC–MS traces of D,D-endopeptidase and D,D-carboxypeptidase activities of Pgp3 with tetra-tri peptide and pentapeptide, respectively. Acceptor strands of crosslinked peptides are shown in gray. The shared structural component between the two substrates is in blue dotted box. **c** TEM analysis of *C. jejuni* strains. *C. jejuni* ATCC 33560 wild-type, Δ*pgp3*, and Δ*pgp3* harboring pPGP3 were negatively stained with 2% (w/v) uranyl acetate and then observed by using TEM. Scale bars (lower left) represent 0.2 μm or 0.5 μm. **d** The effect of Pgp3 deletion on the host cell invasion. Caco-2 epithelial cells were with *C. jejuni* ATCC 33560 wild-type and Δ*pgp3*, and Δ*pgp3*-harboring pPGP3. The numbers of intracellular bacteria were determined 3 h after infection using the gentamicin-protection assay. Error bars represent the means and SEM from three independent experiments. The asterisks show statistical significance (***, $P < 0.0001$). Source data are provided as a Source Data file. **e** Domain architecture of Pgp3 and its stereo ribbon diagram. The N-terminal domain (NTD) and C-terminal domain (CTD) are shown in light green and blue, respectively, whereas the flexible linker and LytM domain are drawn in yellow and orange, respectively. Tartrate molecule is shown as sticks.

1.86 Å to $R_{work}$ and $R_{free}$ values of 17.2% and 19.9%, respectively (Supplementary Table 1). The citrate-bound WT was refined at a resolution of 1.66 Å to $R_{work}$ and $R_{free}$ values of 17.3% and 20.6%, respectively (Supplementary Table 1). When the structure of Pgp3 in complex with tartrate was overlaid with that of the citrate complex, their overall structures were similar with RMSD of 2.95 Å over 256 equivalent $C^{\alpha}$ positions (Figs. 1e, 2a, b). However, both structures showed a large deviation (RMSD 6.98 Å) in two regions (residues Glu99-Phe109 and Thr159-Val164) near the active-site (Fig. 2b). This conformational flexibility is discussed later and has mechanistic implications. We designated the tartrate-bound complex as the open form and the citrate-bound Pgp3 as the closed form, based on the solvent exposure of the active site.

Since crystals of Pgp3 contained one or three subunits in the asymmetric unit of WT and variants (H216A and H247A, both discussed later) depending on the space group, we analyzed the oligomeric states of the WT and variant Pgp3 in solution using size-exclusion chromatography with multi-angle light scattering (SEC-MALS) experiments. The molecular masses of the WT, H216A, and H247A variants were 28.4, 28.9, and 30.0 kDa, respectively, close to the theoretical molecular mass of the Pgp3 monomer (29.4 kDa) (Supplementary Fig. 4). These results indicate that the likely functional state of Pgp3, as measured herein in solution, is monomeric. Interestingly, other structurally

characterized Csd proteins from *H. pylori* may be either a monomer (Csd3) or dimer (Csd1 and Csd2)[5,16].

**Domain architecture and structural comparisons.** A search for overall structural similarities with the full-length Pgp3 using the program DALI failed to reveal any significant matches[24]. For more detailed analysis, we examined each domain individually. The Pgp3 monomer is structurally divided into four regions: the N-terminal domain (NTD, residues 21–95), a flexible linker (residues 96–147), the LytM domain (residues 148–259), and the C-terminal helix (residues 260–273) (Fig. 1e). The strong electron density observed at the metal-binding site of LytM domain was confirmed to be the $Zn^{2+}$ ion by calculating anomalous difference maps using data collected at the zinc absorption edge (1.2824 Å) (Supplementary Fig. 5). The Pgp3 LytM domain contains a highly conserved tetrahedral $Zn^{2+}$-ion-binding motif ($H^{168}xxxD^{172}$ and $HxH^{249}$). The NTD comprises six β-strands arranged in two β-sheets (β2↓-β4↑ and β1↑-β6↓-β5↓-β3↑) (Fig. 1e). According to the DALI search using NTD, the highest similarity was detected in the immunoglobulin domain of mouse leukemia inhibitory factor receptor (mLIFR) (PDB code 2Q7N; chain C; residues 211–286; Z-score 7.2), indicating that the NTD adopts an immunoglobulin-like fold[25,26]. However, the key residues (Lys153, Phe156, and Lys159) for ligand binding in mLIFR were

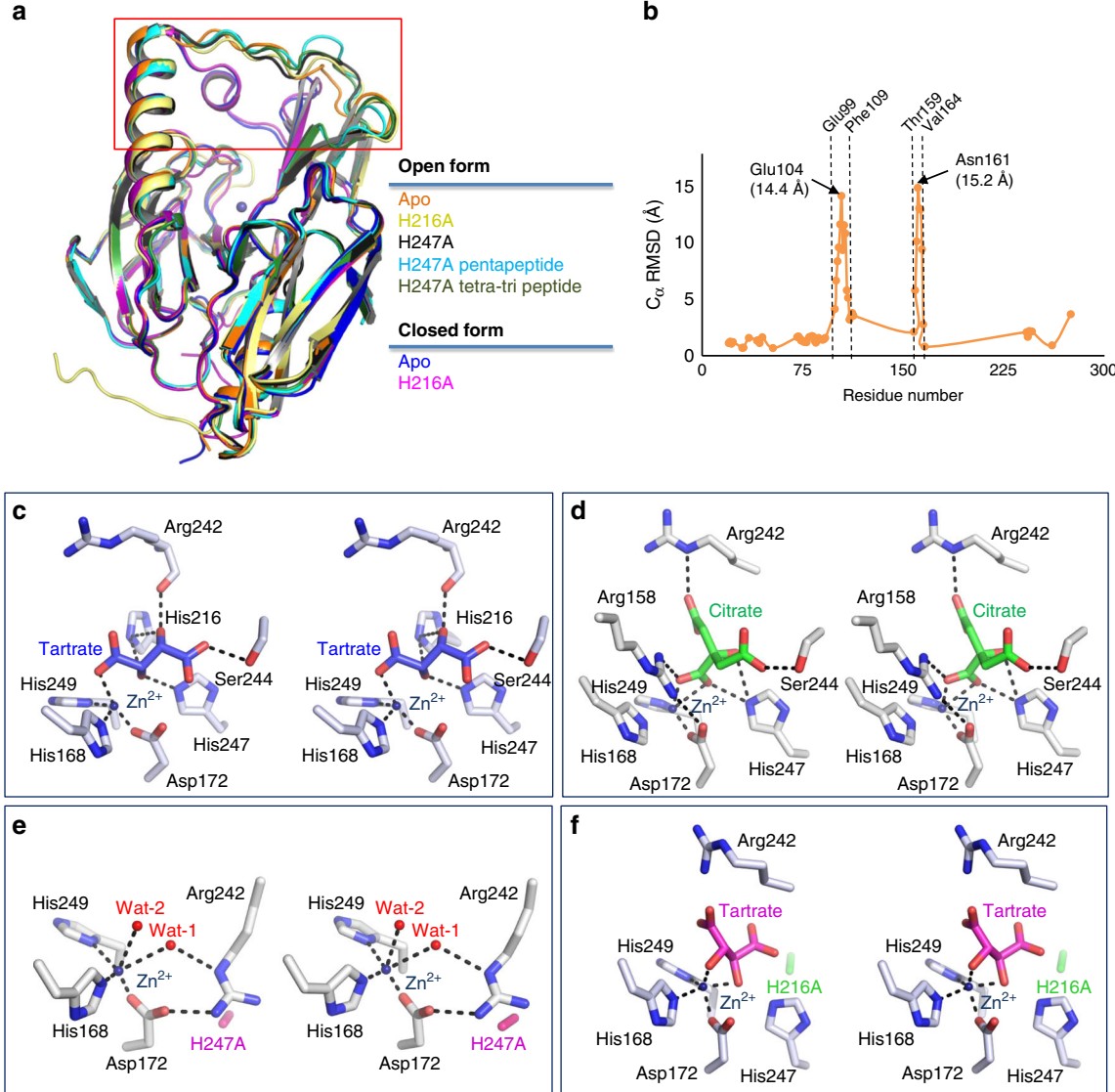

**Fig. 2 Structural comparisons and stereo diagrams of Pgp3 active sites. a** Structural comparisons of seven crystal structures of Pgp3 with open and closed conformations. Superposition of seven crystal structures of Pgp3 (WT open form in orange, H216A mutant open form in light yellow, H247A mutant in black, H247A mutant with pentapeptide in light blue, H247A mutant with tetra-tri peptide in dark green, WT closed form in blue, and H216A mutant closed form in pink). **b** Distance plots between WT Pgp3 in the open form (PDB ID: 6JMX) and corresponding Cα positions of WT Pgp3 in the closed form (PDB ID: 6JMY). **c–f** Stereo diagrams of WT Pgp3 active sites around **c** tartrate (light blue) and **d** citrate (green) molecules. Tetrahedral coordination of $Zn^{2+}$ and hydrogen bonds is represented by black dotted lines, whereas red spheres represent water molecules. **e** Stereo diagrams of H247A mutant (open form). The position of H247A mutation is shown in pink. **f** Stereo diagrams of H216A mutant in complex with tartrate (magenta). The position of H216A mutation is shown in green.

absent in the NTD of Pgp3, suggesting that they are not functionally related[27]. The flexible linker connecting the NTD and LytM domain contains one short β-strand (β7), one α-helix (α1), and one loop (L1). Following the flexible linker, Pgp3 contains the $Zn^{2+}$-coordinated LytM domain with the canonical fold observed in the M23 peptidase family. The LytM domain of Pgp3 comprises a larger, eight-stranded anti-parallel β-sheet (β11↑-β12↓-β13↑-β14↓-β15↑-β16↓-β17↑-β18↓) and a smaller, two-stranded anti-parallel β-sheet (β8↓-β9↑). An anti-parallel arrangement of two β-strands (β8↓-β9↑) is present along with the β7 strand of the flexible linker (Fig. 1e). As expected, the DALI search with the LytM domain of Pgp3 reveals high structural similarity with other LytM domains of peptidoglycan carboxypeptidases (Csd3 from *H. pylori*, PDB code 4RNZ) and the M23 peptidase family (PDB codes 5J1K, 5J1L, and 4ZYB) with high Z-scores (17.8, 16.3, 16.2,

and 15.9, respectively)[25]. The length of the C-terminal helix (residues 260–273) of Pgp3 is shorter than that of two other orthologs in *H. pylori* Csd1 and Csd3. The overall structure of Pgp3 differs remarkably from other structurally characterized Csd proteins (Supplementary Figs. 6 and 7).

**Crystal structures of Pgp3 in complex with its substrates.** Crystals of Pgp3 could only be produced in the presence of either tartrate or citrate molecules. Both molecules bound to the active site of Pgp3 via several hydrogen bonds and salt-bridge interactions between the carboxyl groups of tartrate or citrate and the $Zn^{2+}$ ion (Fig. 2c, d). We decided to mutate residues interacting with tartrate or citrate in order to abrogate binding of the molecules to the active site. By comparing WT Pgp3 with other

M23 family proteins (PDB codes 3IT5 and 3IT7), we noticed that the positions of two conserved water molecules (hereafter referred to as Wat-1 and Wat-2) were substituted by tartrate or citrate oxygen atoms[28]. Specifically, in the tartrate-bound complex, Wat-1 and Wat-2 were displaced by tartrate O3 and O41 atoms, and in the citrate-bound WT Pgp3, they were displaced by citrate O5 and O6 atoms, respectively. Furthermore, in the tartrate-bound complex, the side chain of His216 (NE2 atom) formed a hydrogen bond with the O3 of the tartrate molecule (3.4 Å), and the side chain of His247 (NE2 atom) formed another hydrogen bond with tartrate O3 atom (2.7 Å). In the citrate-bound complex, the side chain of His247 (NE2 atom) formed one hydrogen bond with the citrate O5 atom (2.7 Å), and another hydrogen bond with citrate O7 atom (3.2 Å). Interestingly, Arg158 showed large conformational change, pulling center of mass of β7↑-β8↓-β9↑ sheet by 49° (and 6 Å) in the direction of the active site (compared to the positions in the tartrate-bound complex), and engaged in salt-bridge interactions with the citrate carboxyl group.

The amino acids His216 and His247 are strictly conserved for Wat-1 and Wat-2 coordination in the M23 metallopeptidase family, and for the core of the catalytic machinery. We decided to mutate His216 and His247 individually to alanine in order to abrogate the binding of tartrate or citrate at the active site and impair catalysis too. Indeed, when we solved the crystal structure of the H247A variant, it showed the absence of citrate or tartrate electron densities within the active site and the protein only assumed the open conformation, which was fortuitous for efforts toward solving the structure of the enzyme complexes with substrate analogs (Fig. 2e). In the case of the H216A variant, tartrate was sometimes observed in the closed form crystal structure (Fig. 2f); hence, only the H247A variant was pursued further. The crystallographic data parameters are presented in Supplementary Table 1.

By using the H247A mutant variant, we successfully solved two structures of Pgp3 in complex with a pentapeptide (1, Ac-L-Ala$^1$-$\gamma$-D-Glu$^2$-$m$DAP$^3$-D-Ala$^4$-D-Ala$^5$) and with a crosslinked muramyl tetra-tri peptide (2, NAM-L-Ala$^1$-$\gamma$-D-Glu$^2$-$m$DAP$^3$-D-Ala$^4$-$m$DAP$^3$-$\gamma$-D-Glu$^2$-L-Ala$^1$-NAM). In the presence of the pentapeptide, excellent electron-density map was obtained for the $m$DAP$^3$-D-Ala$^4$-D-Ala$^5$ segment (Fig. 3). In the presence of the crosslinked heptapeptide (tetra-tri peptide), excellent electron density was seen for $m$DAP$^3$-D-Ala$^4$—$m$DAP$^3$. The importance of these two complexes is that the scissile bond in the peptide substrate was seen bound within the active site. The segments of the synthetic substrates that were not seen in the structures argue for mobility at those sites.

Both substrates bound the deep cleft of LytM domain with a 1:1 stoichiometry (Fig. 3). The tetra-tri peptide fragment ($m$DAP$^3$-D-Ala$^4$-$m$DAP$^3$) mainly interacted with Pgp3 via its two terminal $m$DAP moieties. Tyr129, Arg195, and Tyr215 of Pgp3 contributed to the binding by forming several hydrogen-bonding networks. The hydroxyls of Tyr129 and Tyr215 were bound to O7 and O8 atoms of $m$DAP$^3$ (from the donor strand) via hydrogen bonds (3.0 Å and 2.5 Å, respectively). The NH1 atom of Arg195 formed a salt bridge with $m$DAP$^3$ (from the donor strand). The O5 and N1 atoms of the other $m$DAP$^3$ moiety (from the acceptor strand) formed hydrogen bonds with the backbone nitrogen atoms of Val243 and Ser244. These interactions are discussed later in the context of computational simulations with the peptide substrate based on the X-ray structure. When the electrostatic potentials are drawn at the surface of Pgp3, positive electrostatic potentials that are made by the flexible linker (α1 and L1), LytM domain (β8 and β9), and NTD (β4) were widely distributed along the substrate-binding pocket (Fig. 3). Positive charges at the substrate-binding site are complemented by the negatively charged peptidoglycan.

**Structural insights into endolytic and exolytic reactions.** To gain further structural insight into how Pgp3 shows both D,D-carboxypeptidase and D,D-endopeptidase activities, we examined how the $m$DAP$^3$-D-Ala$^4$-D-Ala$^5$ fragment from the pentapeptide and the $m$DAP$^3$-D-Ala$^4$-$m$DAP$^3$ fragment from the tetra-tri peptide is bound at the active site of Pgp3. The critical portions of both peptides with Pgp3 are shared motifs between the two structures. Both peptides contain the common motif, $m$DAP$^3$-D-Ala$^4$, and only the third amino acid is different, i.e., $m$DAP$^3$ vs Ala$^5$ (Fig. 4a). As described above (Fig. 3), Tyr129, Arg195, and Tyr215 of Pgp3 contribute to peptide binding by forming several hydrogen-bonding networks with the first $m$DAP$^3$ moiety of either peptide.

However, the third amino acid of tetra-tri peptide and pentapeptide is different, i.e., $m$DAP$^3$ vs Ala$^5$ (Fig. 4a), thus, we next examined that how the third amino acid of both tetra-tri peptide and pentapeptide is recognized by Pgp3. It is important to note that the third $m$DAP side chain of the tetra-tri peptide encompasses the "D-Ala" segment with the correct stereochemistry corresponding to D-Ala of pentapeptide (Fig. 4b). Interestingly, two more interactions with backbone atoms of Ser244 and Val243 contribute for recognition of the third $m$DAP in tetra-tri peptide (Fig. 3). Moreover, additional tripeptide-NAM-NAM moiety can be linked to N1 atom of the third $m$DAP of tetra-tri peptide because it is solvent-exposed in Pgp3 structure (shown in white-dotted circle in Fig. 4b), suggesting that these structural features are crucial for duality of the endolytic and exolytic reactions of Pgp3.

**The catalytic mechanism.** Our structural studies point to a catalytic mechanism for Pgp3 that is similar to that of other Zn$^{2+}$-dependent hydrolases[27–29] (Fig. 5a). In the uncomplexed H247A variant, two water molecules (Wat-1 and Wat-2) occupy the active site (Fig. 2e). Because Wat-1 and Wat-2 were essentially equidistant from Zn$^{2+}$ (2.9 Å and 2.7 Å, respectively), we could not establish which should be the hydrolytic-water molecule. However, the crystal structure of the H247A variant in complex with the substrate clearly showed that Wat-2 was replaced by the incoming substrates, concluding that Wat-1 is the hydrolytic water that attacks the scissile carbonyl group of substrates. Substrate carbonyl oxygen displaces Wat-2, thereby interacting directly with Zn$^{2+}$ and polarizing the D-Ala$^4$ carbonyl bond. This renders the amide bond susceptible to nucleophilic attack by Wat-1. The distance between Wat-1 and Zn$^{2+}$ decreases from 2.7 Å in the uncomplexed H247A variant to 2.6 Å in the complexed one. The O3 atom of tartrate in WT Pgp3 (open form), which is the position of Wat-1 in H247A variant, is seen interacting with His216 and His247. Since these histidine residues are not involved in Zn$^{2+}$ coordination, we hypothesized that either His216 or His247 might be able to abstract a proton from Wat-1 in promoting the hydrolytic-water molecule for attack at the susceptible substrate carbonyl carbon, en route to the stabilization of the transition state for the reaction within the enzyme oxyanion hole (Fig. 5a). Proton transfer from the histidine general base (His216 or His247) to the departing amide nitrogen facilitates cleavage of the peptide bond to generate a product complex in which the product carboxylate is bound analogously as in the crystallographically observed tartrate (Fig. 5a).

**Computational modeling of Pgp3 with a peptidoglycan.** As discussed above, Pgp3 adopts open and closed conformations due to the large conformational shift of the L1 loop (Fig. 2a). When we superimposed the structure of the H247A variant of Pgp3 in complex with the tetra-tri peptide with that of the WT complex with citrate (closed form), it was clear that the penta- or tetra-tri

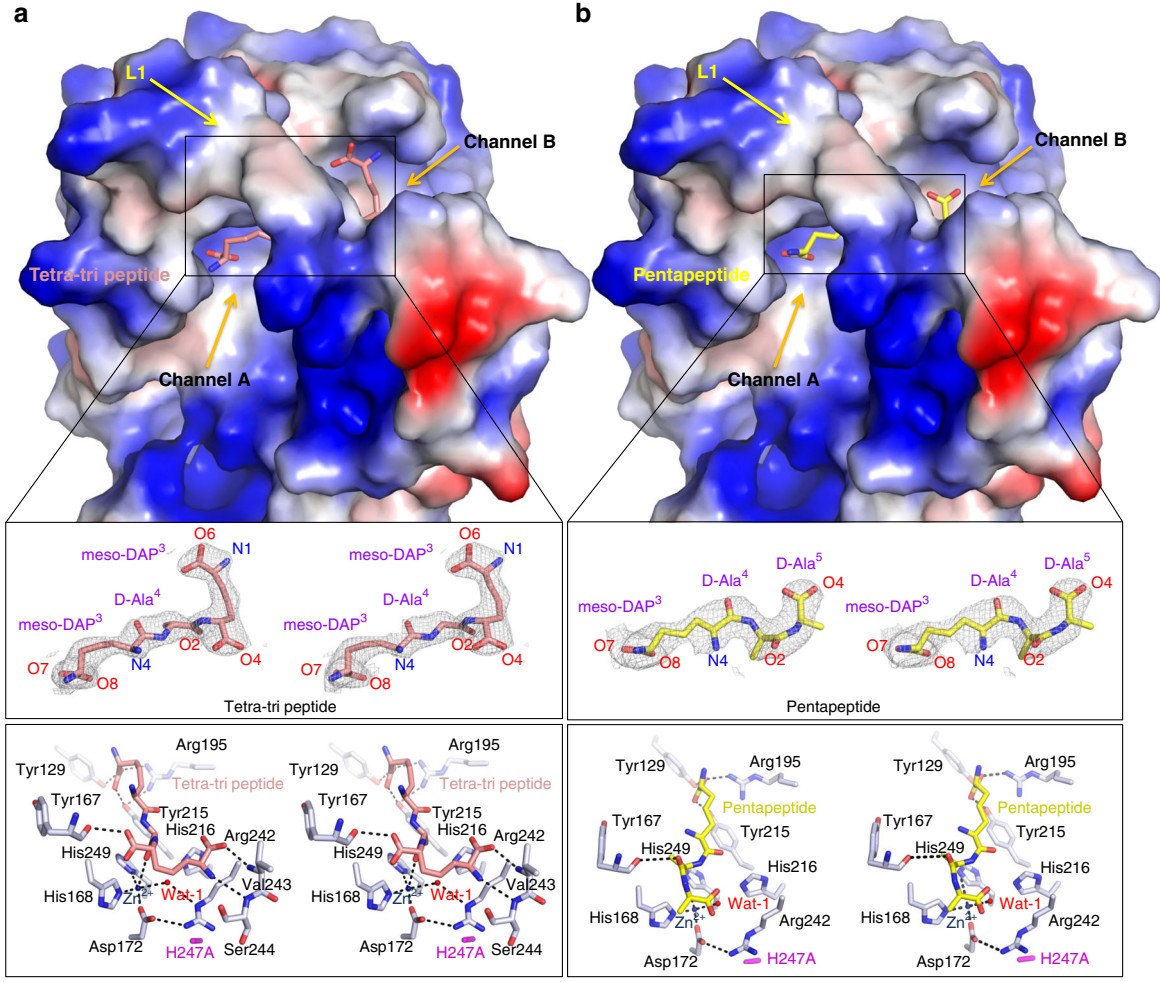

**Fig. 3 Structures of Pgp3 H247A mutant in complex with tetra-tri and pentapeptide. a** Surface representation of the electrostatic potential of Pgp3 in complex with tetra-tri peptide. The bound $m$DAP$^3$-D-Ala$^4$-$m$DAP$^3$ moiety is shown in pink sticks for carbons and expanded with the stereo $2F_O$–$F_C$ electron density map (1.0 $\sigma$). Tetrahedral coordination of Zn$^{2+}$ and hydrogen bonds are represented by black dotted lines, whereas red spheres represent water molecules. **b** Surface representation of the electrostatic potential of Pgp3 in complex with pentapeptide. The bound $m$DAP$^3$-D-Ala$^4$- D-Ala$^5$ moiety is shown in yellow sticks for carbons and expanded with the stereo $2F_O$–$F_C$ electron density map (1.0 $\sigma$).

peptide cannot enter the active site in the closed conformation. The conclusion from our several structures is that catalysis in Pgp3 is regulated by the L1 loop.

The active site resides deep within the LytM domain. Importantly, the N1 and N4 atoms of the two $m$DAP of the tetra-tri peptide are responsible for linkage with the Glu$^2$-Ala$^1$-NAM moiety, which are solvent-exposed through the channels A and B, respectively (Figs. 3, 4). Due to the presence of the L1 loop, the access to the active site by the crosslinked peptidoglycan is prohibited from either the top or the bottom for steric reasons. Intuitively, the substrate could diffuse into the active site only sideways (channel A or B, Fig. 3). The two intact polymeric glycan chains, which define the ends of the crosslinked peptide stem, also prohibit the sideways access of Pgp3. Pgp3 cannot access the intact cell wall of *C. jejuni* without a prior processing of the peptide stem by another enzyme. For example, a different amidase/peptidase may cleave the peptide stem or the product of processing of the saccharide backbone by a lytic transglycosylase. Once this pre-requisite processing takes place, the resultant peptidoglycan, now tethered only to one polymeric glycan chain, can access the active site of Pgp3 from one of the channels. Interestingly, for the helical structure of bacteria, the amidases

AmiA has a crucial role in cell-shape regulation by achieving the cleavage of $N$-acetylmuramoyl-L-alanyl bond (NAM-L-Ala$^1$) and regulating the degree of peptidoglycan crosslinking for the curved morphology[30,31]. The AmiA reaction might serve as the pre-requisite reaction.

To determine which glycan end needs to be processed prior to the reaction of Pgp3, we characterized the approach of the substrate from either channel. We note that a narrow groove defines the surface of Pgp3 near channel A, which could ensconce the polysaccharide backbone from the donor strand in the course of the active-site penetration by the peptide stem (Fig. 5b). We constructed computationally a full-length peptide stem (NAG-NAM(NAG)-L-Ala$^1$-$\gamma$-D-Glu$^2$-$m$DAP$^3$-D-Ala$^4$-$m$DAP$^3$-$\gamma$-D-Glu$^2$-Ala$^1$) within the active site based on the X-ray coordinates, which was submitted to molecular-dynamics simulations. The substrate formed a stable complex with Pgp3 during 250 ns of dynamics simulation, as discerned by its overall RMSD of 0.92 ± 0.23 Å (Fig. 5c). His168, Asp172, His249, and D-Ala$^4$ carbonyl oxygen maintained a tetrahedral geometry with the Zn$^{2+}$ ion during the molecular dynamics (MD) simulation. The scissile amide bond transformed from the *trans* to *cis* configuration rapidly at the beginning of the simulation, revealing that catalysis favored the

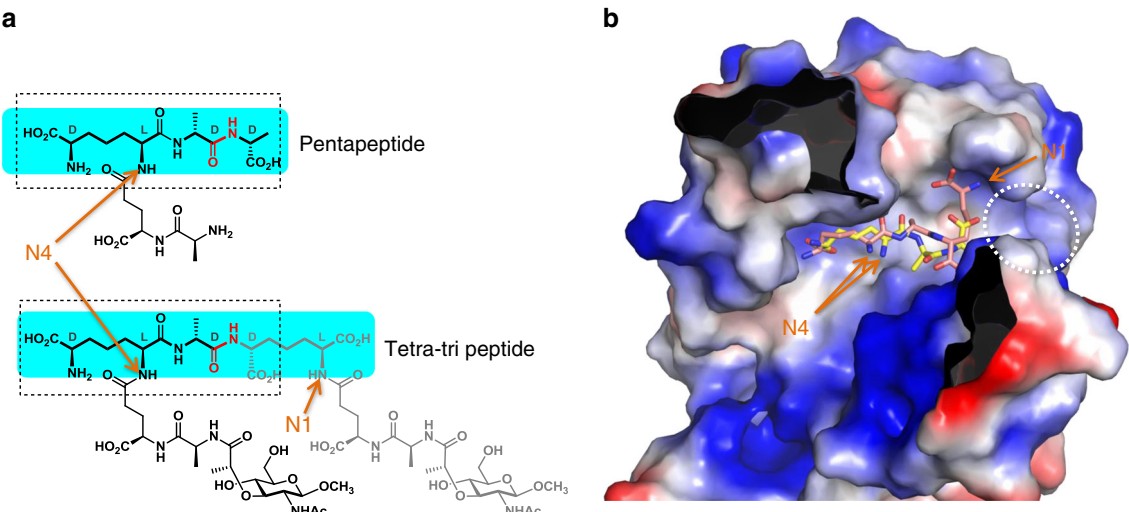

**Fig. 4 Structural comparison of binding modes of penta- and tetra-tri peptides to Pgp3. a** Alignment of the shared structural attributes for the pentapeptide and tetra-tri peptide, identified in the bound ligands. The scissile bonds are depicted in red. The shared structural component between the two substrates is in dotted box with the segments whose electron densities were observed highlighted in the shade boxes in cyan. **b** Surface representation of the electrostatic potential of Pgp3 in complex with pentapeptide (yellow) and tetra-tri peptide (pink). The positions of N1 and N4 atoms of each peptide are indicated by arrows. Solvent-exposed region next to the N1 atom of tetra-tri peptide is indicated by white dotted lines.

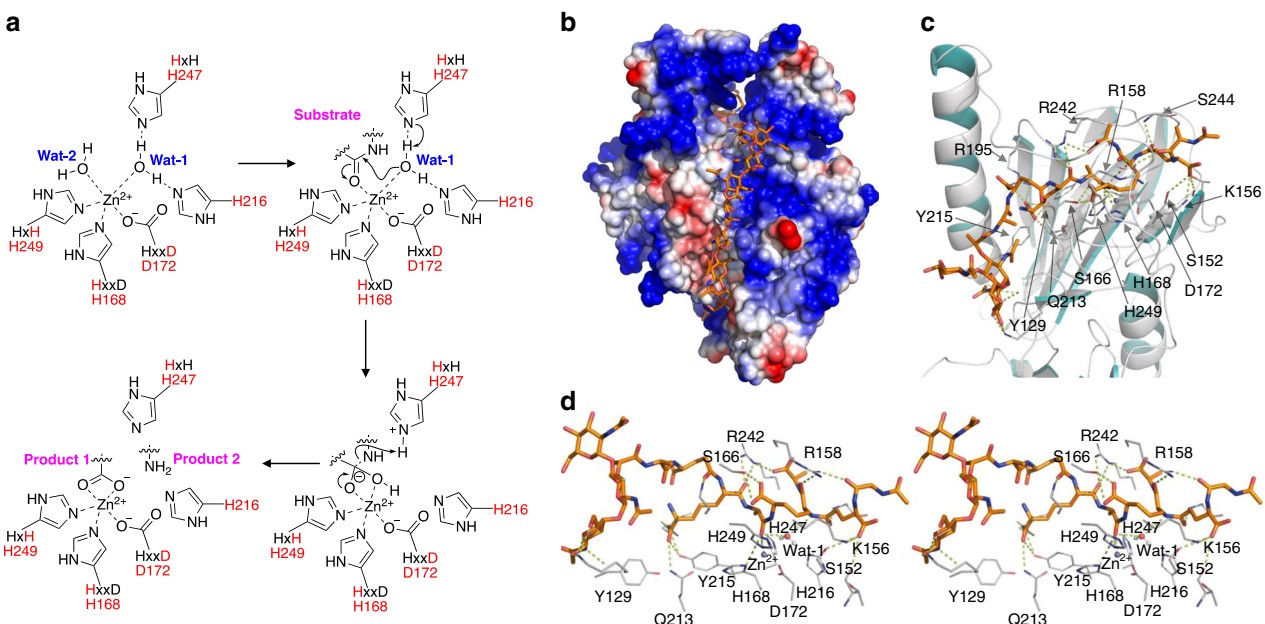

**Fig. 5 Proposed mechanism of Pgp3 and predicted binding mode of peptidoglycan. a** The proposed catalytic mechanism for turnover by Pgp3. **b** A polymeric strand of peptidoglycan is shown ensconced in a groove leading to the active site. **c** Binding mode and interactions of the modeled peptidoglycan substrate in a snapshot of the 250-ns MD trajectory. Tyr129, Arg195, Gln213, and Tyr215 form significant interactions with the carboxyl group of $m$DAP[3]; Ser152 and Lys156 interact with carboxyl group of $\gamma$-D-Glu[2]; Arg158 interacts with $\gamma$-D-Glu[2]; Ser166 and Arg242 interact with carboxyl group of $m$DAP[3] and Ser244 interacts with $\gamma$-D-Glu[2]. **d** Stereo diagrams of the active-site bound peptidoglycan.

*cis* configuration of amide bond. Wat-1 also formed strong hydrogen bonds with His216 and His247. During MD, Wat-1 was properly aligned to transfer a proton to either His216 or His247 (average distance from Wat-1 O atom, $2.76 \pm 0.04$ Å and $2.71 \pm 0.04$ Å, respectively) prior to nucleophilic attack on the polarized carbonyl carbon of the substrate (average distance from Wat-1 O atom of $3.20 \pm 0.04$ Å). Tyr129, Arg195, and Tyr215 also consistently maintained hydrogen bonds and salt-bridge interactions with carboxyl group of $m$DAP[3] during MD simulation

(occupancies of 15%, 35%, and 36%, respectively). The carboxylate of the other $m$DAP[3] and of D-Ala[4] were seen interacting with the highly flexible Arg242 and Ser166 through salt-bridges and hydrogen bonds, respectively. One $\gamma$-D-Glu[2] could be seen interacting with Ser152, Lys156, and Ser244, while the other $\gamma$-D-Glu[2] formed intramolecular salt-bridge interactions with $m$DAP[3] (Fig. 5d). The carbonyl group of L-Ala[1] formed a hydrogen bond with Arg158, which is present on the β8 strand (part of β7↑-β8↓-β9↑ near L1 linker loop region). In the pentapeptide-bound

complex, Arg158 might form salt-bridge interactions with D-Ala[5]. Upon substrate binding, Arg158 extended into the active site and pulled L1 loop toward the active site. This indicates that the active site tends to close off after substrate binding. That is to say that the apo enzyme would sample both closed and open conformations. For catalysis to proceed, the active site needs to open for the substrate to bind, but on binding, the loop closes again to enable catalysis, which then would release the two products from both channels A and B. This gives rise to the closed apo enzyme (seen in our structure), which is ready now to repeat the process.

We also analyzed the collective motions of Pgp3 with the help of temperature factors and elastic-network models. The L1 loop and loops between β12↓-β13↑ and β17↑-β18↓ play a role in the conformational dynamics. We also fine-tuned the conformational dynamics upon substrate binding, which we discerned in equilibrium dynamics, using guided sampling to expedite the conformational transition. We used temperature-Replica Exchange MD (tREMD) and Targeted MD (TMD). While tREMD utilizes temperature to cross the high-energy barrier between conformations, TMD uses RMSD between two pre-defined end states to drive the conformational change. Whereas tREMD sampled some of the conformational states in transition from one state to the other, TMD was more conclusive (smallest RMSD between the linker L1 in the closed conformation, and any MD snapshot in TMD and tREMD were 3.20 Å and 2.28 Å, respectively). Arg158 played a key role in pulling the β-sheet near L1 loop toward the substrate, which led to relaxing of the taut L1 loop, which eventually reorganized into a single-turn helix. A plot depicting RMSD between open and closed states during REMD and TMD and a Supplementary movie depicting the conformational dynamics is shown in Supplementary Fig. 8 and Supplementary Movie 1.

## Discussion

We have described herein the reactions and the structure of Pgp3, a Csd protein of *C. jejuni*. This enzyme turns over the cell-wall peptidoglycan by both the D,D-endopeptidase and D,D-carboxypeptidase activities, as documented by mass spectrometric characterization of the requisite reactions with synthetic cell-wall fragments. The high-resolution structures and molecular dynamics reveal an elaborate process of conformational change that samples the enzyme between active-site closed and open states. Furthermore, based on the topographical construction of the active site, our work reveals that the peptidoglycan substrate for this enzyme can be tethered to the polymeric cell-wall saccharide backbone only on one strand (and not two). These are the structural and mechanistic insights into an important enzyme that regulates the shape and virulence of *C. jejuni*.

## Methods

**Protein expression and purification.** The gene encoding residues Glu21–Gln273 Pgp3 from *C. jejuni* (strain ATCC 33560) was PCR-amplified using primer sets, Pgp3-NdeI-F and Pgp3-XhoI-R (Supplementary Table 2) and cloned into the expression vector pET-21a (+) (Novagen) using *NdeI* and *XhoI* restriction enzymes, producing the recombinant Pgp3 protein with the C-terminal His₆-tag (LEHHHHHH). The Pgp3 protein was expressed in *E. coli* Rosetta2 (DE3) cells (Novagen) induced with 0.5 mM isopropyl *β*-d-thiogalactopyranoside (IPTG) at 30 °C for 16 h following growth to mid-log phase at 37 °C. The harvested cell pellet was resuspended in a lysis buffer containing 20 mM Tris-HCl pH 7.9, 500 mM sodium chloride, 5 mM imidazole, 10% glycerol, and 1 mM phenylmethylsulfonyl fluoride (PMSF) and was lysed by sonication. After centrifugation at 36,000 × *g* for 1 h at 4 °C, the cell debris was discarded, and the supernatant was applied to an affinity chromatography column of HiTrap chelating HP (GE Healthcare), which was previously equilibrated with the lysis buffer. The column was washed with lysis buffer containing 25 mM imidazole and was eluted with a linear gradient from 25 to 500 mM imidazole. The protein was eluted at 150−200 mM and further purified by gel filtration on a HiLoad 16/60 Superdex 200 prep-grade column (GE Healthcare), which was previously equilibrated with 20 mM HEPES pH 7.5, 200 mM NaCl, and 1 mM DTT. Peak fractions containing the Pgp3 protein were

pooled and concentrated to 10 mg mL⁻¹ for crystallization. The mutants of Pgp3 (H216A and H247A) were generated by site-directed mutagenesis using primer sets, Pgp3-H216A-F, Pgp3-H216A-R, Pgp3-H247A-F, and Pgp3-H247A-R (Supplementary Table 2), and were purified as described for WT Pgp3.

**Crystallization and data collection.** We determined two structures of WT Pgp3, i.e., tartrate- and citrate-bound forms. Crystals were grown at 296 K by the sitting-drop vapor-diffusion method using the Mosquito robotic system (TTP Labtech). For the tartrate-bound form, the sitting drop was prepared by mixing equal volumes of the protein solution (10 mg mL⁻¹, 0.2 μL) and reservoir solution (0.2 μL) containing 200 mM potassium sodium tartrate, 100 mM tri-sodium citrate at pH 5.6, and 2 M ammonium sulfate. Crystals of tartrate-bound Pgp3 were cryoprotected in the reservoir solution supplemented with 20% (v/v) glycerol and were flash-frozen in a nitrogen gas stream at 100 K. For the citrate-bound form, the sitting drop was prepared by mixing equal volumes of the protein solution (10 mg mL⁻¹, 0.2 μL) and reservoir solution (0.2 μL) containing 10% 2-propanol, 100 mM citrate at pH 5.5, and 20% PEG4000. Crystals of citrate-bound Pgp3 were cryoprotected in the reservoir solution supplemented with 10% (v/v) glycerol and were flash-frozen in a nitrogen gas stream at 100 K. Native data for the tartrate-bound and citrate-bound Pgp3 were collected at 1.86 Å and 1.66 Å resolutions, respectively, using the ADSC Q315r CCD detector at the beamline BL-5C of Pohang Light Source (PLS), Pohang, Korea (Supplementary Table 1).

Crystals of H216A variant in apo-form and tartrate-bound form were grown at 296 K by sitting-drop vapor diffusion method using the Mosquito robotic system (TTP Labtech). For the crystal of the H216A apo-form, sitting drop was prepared by mixing equal volumes of the protein solution (10 mg mL⁻¹, 0.2 μL) and reservoir solution (0.2 μL) containing 200 mM potassium sodium tartrate tetrahydrate, 100 mM sodium citrate tribasic dehydrate at pH 5.6, and 2.0 M ammonium sulfate. The crystal of the H216A tartrate-bound form was prepared by mixing equal volumes of the protein solution (10 mg mL⁻¹, 0.2 μL) and reservoir solution (0.2 μL) containing 1.1 M malonic acid, 150 mM ammonium citrate tribasic, 72 mM succinic acid, 180 mM DL-malic acid, 240 mM sodium acetate, 300 mM sodium formate, and 96 mM ammonium tartrate dibasic at pH 7.0. Both crystals were soaked in Paratone-N (Hampton Research, Aliso Viejo, CA, USA) before being flash-frozen in a nitrogen stream at 100 K. Native data were collected to 2.10 Å and 2.04 Å resolutions for the apo-form and tartrate-bound form, respectively, using the beamline BL-5C of PLS (Supplementary Table 1).

For the Pgp3 H247A variant crystal, sitting drop was prepared by mixing equal volumes of the protein solution (10 mg mL⁻¹, 0.2 μL) and reservoir solution (0.2 μL) containing 100 mM sodium citrate tribasic dihydrate at pH 4.5, 200 mM potassium sodium tartrate tetrahydrate, and 2.0 M ammonium sulfate. For the penta or tetra-tri peptide complex, H247A protein (10 mg mL⁻¹) in 20 mM Tris-HCl pH 8.0, 200 mM NaCl, and 70 mM peptides in 150 mM Tris-HCl pH 8.0 were mixed in a 1:10 molar ratio and incubated at 277 K for 1 h. Crystals were grown at 296 K by the sitting-drop vapor diffusion method by mixing equal volumes of the protein–peptide mixture (0.2 μL) and reservoir solution (0.2 μL) containing 1.8 M ammonium citrate tribasic at pH 7.0. Crystals reached maximum size within 2–3 days. They were cryoprotected in the reservoir solution supplemented with 30% glycerol and were flash-frozen in a nitrogen gas stream at 100 K. X-ray diffraction data were collected in 1° oscillations at beamline BL-7A of PLS (Supplementary Table 1). Raw X-ray diffraction data for all crystals mentioned above were processed and scaled using the program suit HKL2000 (ref. [32]). Data collection statistics are summarized in Supplementary Table 1.

**Structure determination and refinement.** Two WT structures of tartrate- and citrate-bound Pgp3 were determined by molecular replacement utilizing the program MOLREP, with putative peptidase M23 from *P. aeruginosa* (PDB code 2HSI) as a search model (unpublished data)[33]. The peptidase M23 (residues Arg68–Gln281) shows 34% sequence identity with Pgp3 (residues Glu57–Asn269). Manual model building was performed using the program COOT and models were refined with the program REFMAC5, including the bulk solvent correction[34,35]. In total, 5% of the data were randomly set aside as test data for calculating $R_{free}$[36]. The stereochemistry of the refined models was assessed using MolProbity[37]. Crystallographic and refinement statistics are summarized in Supplementary Table 1.

The structures of the H216A, H247A, and H247A variants in complex with two peptides were also solved by the molecular replacement method utilizing the monomer model (chain A) of WT Pgp3. Subsequent manual model building was carried out using the COOT program and restrained refinement was performed using the REFMAC5 program[35]. Several rounds of model building, simulated annealing, positional refinement, and individual *B*-factor refinement were performed. Supplementary Table 1 lists the refinement statistics. Atomic coordinates and structure factors of eight crystal structures of Pgp3: (i) tartrate-bound form, (ii) citrate-bound form (two structures), (iii) H216A variant, (iv) H216A variant with tartrate, (v) H247A variant, (vi) H247A variant with pentapeptide (L-Ala¹-γ-D-Glu²-*m*DAP³-D-Ala⁴-D-Ala⁵), (vii) H247A variant with crosslinked muramyl tetra-tri peptide (NAM-L-Ala¹-γ-D-Glu²-*m*DAP³-D-Ala⁴-*m*DAP³-γ-D-Glu²-Ala¹) have been deposited in PDB (PDB ID codes 6JMX, 6JMY, 6KV1, 6JN8, 6JN7, 6JMZ, 6JN1, and 6JN0).

**SEC-MALS**. SEC-MALS experiments for WT and mutants (H216A and H247A) of Pgp3 were performed using an FPLC system (GE Healthcare) connected to a Wyatt MiniDAWN TREOS MALS instrument and a Wyatt Optilab rEX differential refractometer. A Superdex 200 10/300 GL (GE Healthcare) gel filtration column pre-equilibrated with buffer containing 20 mM Tris-HCl pH 8.0 and 200 mM NaCl was normalized using ovalbumin. Proteins were injected at a flow rate of 0.4 mL min$^{-1}$. Data were analyzed using the Zimm model for static light-scattering data fitting and graphs were constructed using EASI graph with a UV peak in the ASTRA V software (Wyatt).

**Computational modeling**. We tested the conformational transition between two structures of WT Pgp3 upon substrate binding using computational methods. Three-dimensional crystal structures of WT and mutant Pgp3 (H216A and H247A) were superimposed in PyMol v1.7 and the coordinates of the co-crystallized ligand in H247A mutant, i.e. tetra-tri peptide ligand, were imported inside WT Pgp3. The ligand (mDAP$^3$-D-Ala$^4$-mDAP$^3$) position was used to append the peptide chains (L-Ala$^1$-γ-D-Glu$^2$ and γ-D-Glu$^2$-L-Ala$^1$) on either ends to complete the model of crosslinked peptide chain (L-Ala$^1$-γ-D-Glu$^2$-mDAP$^3$-D-Ala$^4$-mDAP$^3$-γ-D-Glu$^2$-L-Ala$^1$). In the lack of concrete evidence about the nature of Pgp3 substrate (murein sacculus or muropeptides) and considering the relevance of AmiA in *C. jejuni*, the glycan residues (NAG-NAM-NAG) were attached only on one end of peptide chain in order to leave one side open to facilitate access of Pgp3. Ultimately, as mentioned a priori, the glycan residues are solvent exposed and impact peptide-binding minimally. Therefore, even our general model can be fitted to all plausible Pgp3 substrates (murein sacculus that are modified by AmiA or NAG-NAM, NAG-anhydroNAM, NAM muropeptides on either or both ends). For modeling WT complex, the active-site Zn$^{2+}$ and its tetrahedrally coordinated residues, His168, Asp172, His249, and peptidoglycan substrate were treated with QM/MM. DFT level of theory with B3LYP functional and 6-31G* basis set was used to optimize geometry of residues and perform charge refitting with RESP. A bonded-coordination model between Zn$^{2+}$, NE atom of His168, OD1 atom of Asp172, ND atom of His249, and polarized carbonyl oxygen of substrate's -D-Ala$^4$-mDAP$^3$- amide bond was constructed using MCPB module of AmberTools17. Additionally, His216 and His247 were restrained along with a water molecule in Wat-1 location, unless otherwise specified. The WT complex files were prepared with Amber forcefield ff14SB and waters were treated with either TIP4P or OPC models inside a rectangular box with 10 Å buffer. All MD simulations were carried out using GPU-supported Amber18 MD package. The equilibrium dynamics of WT complex was conducted with constant temperature and pressure (NPT) and periodic boundary conditions. Particle Ewald Mesh (PME) was used to calculate long-range electrostatic interactions and the non-bonded cutoff was set to 10.0 Å. SHAKE was applied to constrain bonds involving hydrogen and temperature was maintained at 300 K using Langevin thermostat with a collision frequency of 5.0 ps$^{-1}$. An average pressure of 1 atm was maintained with isotropic scaling and 2 ps relaxation time. The integration time step of 2 fs was used, unless otherwise specified. The complex was first minimized, and the temperature was raised from 0 K to 300 K in 40,000 steps gradually. A brief equilibration of 500 ps was performed before starting production run of 500 ns. Three additional 10-ns apo-WT MD simulations, (i) restrained protein, unrestrained water, (ii) T-REMD, and (iii) TMD were also performed to study the apo-state Zn$^{2+}$ coordination character and conformational dynamics between open and closed states, respectively.

In REMD, 60 replicas were selected depending on the square root of the number of atoms in the system and the temperature was varied between 270 and 600 K in geometric progression. The chirality restraints were imposed on the backbone to avoid non-physical rotation of peptide bonds at high temperatures. The REMD was carried out using constant temperature, constant volume (NVT) dynamics with 2 fs time step and exchange between replicas was allowed every 1000 steps. In TMD, an additional energy term was added based on the RMSD between L1 loop residues, in the open and closed apo-states of WT Pgp3. The energy term has a harmonic nature; $V = \frac{1}{2} \times k \times (RMSD(t) - RMSD_0(t))^2$, where $k$ has a force constant value of 5 kcal mol$^{-1}$ Å$^2$ (calibrated after several trials), RMSD(t) is the RMSD at simulation time $t$ between current snapshot and reference structure and RMSD$_0(t)$ is the prescribed RMSD at simulation time $t$ between current snapshot and reference structure. The starting RMSD was set to 6.983 Å and then it was decreased gradually to 0 Å in 5 ns and kept at 0 Å for next 5 ns. All the clustering treatments were done by CPPTRAJ in AmberTools17 using DBSCAN clustering method.

**Reaction of Pgp3 with synthetic peptidoglycans**. The reactions of Pgp3 with seven synthetic peptidoglycan substrates (2–8, Supplementary Fig. 2 for chemical structures) were carried out in 50 mM Tris, pH 7.5 at 23 °C and were stopped by the addition of trifluoroacetic acid. The resultant mixture was analyzed by LC/MS whose conditions were reported previously[21]. Seven synthetic substrates and five product standards were synthesized by the methodology reported previously[21–23,38]. Reactions are summarized in Supplementary Fig. 2.

**Bacterial strains and culture conditions**. *C. jejuni* strain (ATCC 33560) was grown at 42 °C in Mueller-Hinton (MH) media (Oxoid, Canada) under a micro-aerobic

condition (5% O$_2$, 10% CO$_2$, 85% N$_2$). Occasionally, MH media were supplemented with kanamycin (50 μg mL$^{-1}$) and/or chloramphenicol (12.5 μg mL$^{-1}$). *E. coli* DH5α (Novagen) harboring plasmids was grown at 37 °C in Luria-Bertani (LB) media (Difco, US) that were supplemented with carbenicillin (100 μg mL$^{-1}$), kanamycin (50 μg mL$^{-1}$), or chloramphenicol (12.5 μg mL$^{-1}$), where required.

**Generation and complementation of the *pgp3* mutant**. The *pgp3* gene and its flanking region in *C. jejuni* ATCC 33560 were amplified by PCR using primer sets, pgp3-PstI-F and pgp3-SalI-R (Supplementary Table 2). After digestion with *PstI* and *SalI*, the fragment was ligated into pUC19 that had been digested with the same enzymes. The ligation product was transformed into *E. coli* DH5α competent cells, and then the whole plasmid except for part of the target gene was amplified by inverse PCR with pgp3-inverse-F and pgp3-inverse-R (Supplementary Table 2). Purified inverse PCR products were ligated to a kanamycin-resistant cassette amplified with Kan-F and Kan-R from pMW10 (Supplementary Table 2), and the resulting suicide vector was transformed into *E. coli* DH5α. The generated suicide vector was electroporated into *C. jejuni* ATCC 33560, and mutants were selected on MH agar plates supplemented with kanamycin (50 μg mL$^{-1}$). The *pgp3* mutation was confirmed by PCR and sequencing.

The complemented strain was created by amplifying coding regions of *pgp3* along with its potential promoter region by PCR using pgp3-comple-F and pgp3-comple-R (Supplementary Table 2). After digestion with *XbaI*, the fragment was cloned into *XbaI* digested pFMBcomCM[39]. The pFMBcomCM::*pgp3* (pPGP3) was electroporated into *pgp3*-mutant strain, and complemented strains were selected on MH agar plates supplemented with kanamycin (50 μg mL$^{-1}$) and chloramphenicol (12.5 μg mL$^{-1}$).

**TEM analysis**. *C. jejuni* strains were harvested from MH agar plates after over-night growth and were resuspended in water. After being negatively stained with 2% uranyl acetate for 10 s on formvar/carbon copper grids (200 mesh), morphology was examined with an energy-filtering transmission microscope (EF-TEM; Libra 120, Germany) at voltage of 120 kV.

**Cell line, media, and culture conditions**. Human intestinal epithelial Caco-2 cell lines were obtained from American Type Culture Collection (Manassas, VA). Caco-2 cells were maintained in Dulbecco's modified eagle medium (DMEM; ATCC) with 10% fetal bovine serum (FBS; Invitrogen), and incubated at 37 °C under 5% CO$_2$. The cell viability was determined with trypan blue staining.

**Invasion assay**. Prior to bacterial infection, a monolayer of $1 \times 10^5$ Caco-2 cells was prepared in a 24-well tissue culture plate[40]. Overnight cultures of *C. jejuni* strains were harvested from MH agar plates and suspended in MH broth to an OD$_{600}$ of 0.1. After culturing 6 h at 42 °C with shaking under a micro-aerobic condition, the bacterial suspension was diluted in Dulbecco's modified eagle medium (DMEM; ATCC) with 10% fetal bovine serum (FBS; Invitrogen), and then added onto the cell monolayer at a multiplicity of infection (MOI) of 100 (ref. [40]). After a 3-h incubation, the wells were washed three times with pre-warmed PBS and then incubated for 3 h with the fresh pre-warmed medium supplemented with 250 μg mL$^{-1}$ of gentamicin to kill the extracellular bacteria[40]. Subsequently, the wells were washed three times with PBS, lysed in 0.1% Triton X-100 for 15 min, and then serially diluted in PBS[40]. A dilution of the suspension was plated on MH agar plates to enumerate the CFU. All invasion assays were done in triplicate wells.

**Statistical analysis**. Statistical analysis was performed using the GraphPad Prism 5.01 software. All results were analyzed by Student's unpaired $t$ test. The data are presented as means and standard errors of the mean (SEM).

**Reaction of Pgp3 with sacculus**. The sacculus of *E. coli* K12 was isolated by the general procedure described previously[41]. The reaction of Pgp3 with sacculus was carried out in 50 mM Tris, pH 7.5 at 37 °C. After 20 h of incubation, the reactions were stopped by boiling the mixture for 5 min. The resultant suspension was treated with mutanolysin for 20 h at 37 °C. The reactions were stopped by boiling the mixture for 5 min and reduction of the reducing ends by sodium borohydride. After centrifugation ($17,000 \times g$ for 10 min), the supernatants of the resultant mixtures were subjected to LC/MS analysis, whose conditions were reported previously[38]. Reaction outcomes are summarized in Supplementary Fig. 3.

**Reporting summary**. Further information on research design is available in the Nature Research Reporting Summary linked to this article.

## Data availability

Atomic coordinates and structure factors of the eight crystal structures of Pgp3 have been deposited in PDB (PDB ID codes 6JMX (https://www.rcsb.org/structure/6JMX), 6JMY (https://www.rcsb.org/structure/6JMY), 6KV1 (https://www.rcsb.org/structure/6KV1), 6JN8 (https://www.rcsb.org/structure/6JN8), 6JN7 (https://www.rcsb.org/structure/6JN7), 6JMZ (https://www.rcsb.org/structure/6JMZ), 6JN1 (https://www.rcsb.org/structure/6JN1), and 6JN0 (https://www.rcsb.org/structure/6JN0). The source data

underlying Fig. 1d are provided as a Source Data file. Other data are available from the corresponding authors on reasonable request.

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

## Acknowledgements

The authors thank the staff of beamlines BL-5C and BL-7A at the Pohang Light Source, and beamline BL44XU at Spring-8 for their assistance during X-ray experiments. The authors thank Yuri Choi, Hyunjun Yoon, and Yoonyoung Heo for assistance during X-ray experiments. This study was supported by a grant from the National Research Foundation (NRF) of Korea funded by the Korean government (2015R1A5A1008958 and 2018R1A2B2008142) and a grant from the Korea CCS R&D Center (KCRC; 2014M1A8A1049296) to H.H.L; a grant from Korea Ministry of Science, ICT and Future Planning, National Research Foundation (NRF) of Korea (2013R1A2A1A05067303) to S.W.S.; a grant from the National Research Foundation (NRF) of Korea funded by the Korean government (2018R1D1A1B07040808) to Y.H.J. The work in the USA was supported by grants from the National Institutes of Health (GM131685 and GM61629 to S.M.).

## Author contributions

K.M., D.R.A., S.M., S.W.S. and H.H.L. conceived and designed the experiments. K.M., D.R.A. and H.-J.Y. solved crystal structures. N.R. performed computations. M.L. performed the turnover experiments, mass spectrometry, and synthesized the substrates. D.H., J.S.P. and B.M.K. contributed to the syntheses of substrates. J.K. and S.R. performed *C. jejuni* morphology and invasion studies. K.M., N.R., M.L., S.M., S.W.S. and H.H.L. analyzed the data and wrote the manuscript. S.M., S.W.S. and H.H.L. directed the teams. All authors edited the manuscript.

## Competing interests

The authors declare no competing interests.
