## [Peer Review File · Nature Communications]

Reviewers' comments:

Reviewer #1 (Remarks to the Author):

The manuscript reports several crystal structures of a new peptidoglycan hydrolase, Pgp3, from *Campylobacter jejuni*, and shows that the enzyme has DD-endopeptidase and DD-carboxypeptidase activity using synthetic peptidoglycan fragments as substrates. Pgp3 was identified by its sequence similarity to known members of the LytM type metallopeptidase family, for example the Csd peptidoglycan hydrolases (Csd1, Csd2, Csd4 and Csd6) from *Helicobacter pylori*, and Psp1 and Psp2 from *Campylobacter jejuni*. These enzymes are required for maintaining the helical cell shape in both pathogens. The current manuscript presents the crystal structure of Psp3 alone and of an inactive version with bound peptidoglycan fragments. The authors present convincing data on the importance of the catalytic Zn, which is expected based on the published structures of homologous hydrolases. They conclude that Pgp3 turns over peptidoglycan and that it is important to regulate the shape and virulence of *Campylobacter jejuni*. Whilst this is a solid structural work in my view the manuscript has major problems. In particular, they draw conclusions that are not supported by experimental data. There are many examples of proteins that don't have the same cellular function as an homolog, hence the function of a novel protein must be carefully determined and cannot be deduced from the function of its homologs.

Major concerns:

1. Title, Abstract and Concluding Remarks. The authors conclude that Pgp3 is "a cell shape determining protein" (line 340) and "an important enzyme that regulates the shape of ... *C. jejuni*." (line 349). This conclusion is not supported by experimental data. Hence, the authors should perform the following experiments:

1.1 Analyse the cell shape of a mutant strain that lacks the *pgp3* gene and assess if there are any cell shape defects.

1.2 Should the mutant strain have cell shape defects, complement the mutant strain by ectopic expression of the *pgp3* gene and test if the shape of wild-type cells is restored.

2. Concluding remarks (line 350): The authors conclude that Pgp3 is "an important enzyme that regulates the ... virulence of ... *C. jejuni*". This conclusion is not supported by experimental data. Hence, the authors should perform the following experiments:

1.1 Analyse the virulence of a mutant strain that lacks the *pgp3* gene, to test if there is any change in virulence compared to wild-type.

1.2 If the mutant strain shows altered virulence, complement the mutant strain by ectopic expression of the *pgp3* gene and test if virulence is restored.

3. The authors also claim that "This enzyme turns over cell-wall peptidoglycan ...". However, they have shown activity only with soluble peptidoglycan fragments, and not with cell wall peptidoglycan. To test if Pgp3 is capable of digesting cell wall peptidoglycan they should:

3.1. test the activity against peptidoglycan from *Campylobacter*.

3.2. test the activity against peptidoglycan in the presence of an MurNAc-L-Ala peptidoglycan amidase. This experiment is crucial to test the model presented in Supplemental Figure 7.

3.3 compare the peptidoglycan composition of a *pgp3* mutant strain with that of wild-type.

Minor point:

4. I would not use the term "bi-functional enzyme" (line 343) to characterize an enzyme with DD-endopeptidase and DD-carboxypeptidase activity, because both activities involve the same active site and similar substrates. A bifunctional enzyme would rather be one that has two different active sites/domains catalysing different reactions.

5. line 342. The synthetic peptidoglycan fragments used were not perfectly "authentic", as they were methylated at the MurNAc residue, which would not occur in natural peptidoglycan.

6. line 203 and elsewhere. It should be added that the Ala1 residue has L-configuration.

Reviewer #2 (Remarks to the Author):

This manuscript describes an interesting, important and topical area of science; the maintenance of bacterial viability and identification of potential targets for new antibiotics.

The study is truly multidisciplinary with structural biology and molecular simulations playing key roles.

I am only able to comment on the simulations as an expert; regarding the X-ray structures I can comment as a knowledgeable inexpert!

I think overall the study is of high quality and the data are interesting.

I think the simulation models and methods are appropriate. However I am unclear as to how many independent simulations of each system were conducted. I can only assume that this is more than 2, if not then it should be to demonstrate reproducibility. I have no other real concerns.

Minor comments.

Line 173 'Both tightly bonds to the active site' - please rephrase this to make it clear you are referring to the shape fit of the molecules here and not energetics.

Line 188 'pulling the β 7- β 8- β 9 sheet by 49 ° - a distance measurement might be useful here too?

252-253 'indicating that Wat-1 is the hydrolytic water' please rephrase this - it only suggests that it is likely to be the key water, it is not conclusive, please make this clear.

Supporting information

'Protein expression and purification' <-- correct typographical error

Line 76 (gold standard of water models) <-- please remove this as accuracy of water models is very much dependent upon the quantities being measured.

For the REMD simulations please explain why the lowest temperature was chosen to be 270 K when the other, traditional MD simulations are performed at 300 K.

Reviewer #3 (Remarks to the Author):

Min et al., report several high-resolution X-ray structures and the biochemical characterization of the Pgp3 peptidase from *C. jejuni*. This bacterium is a human pathogen with a high incidence in term of associated guts disease. The reason for the atypical helical shape of this bacterium is not completely understood. Identifying key player(s) responsible for this shape and understanding their role could potentially pave the way to identify new drug target.

In this study, the authors solved numerous crystal structure of the Pgp3 protein in its apo and closed forms and in the presence of peptidoglycan synthetic fragments. Thanks to these structures a very interesting mechanism of peptidoglycan recognition and hydrolysis have been proposed. Rare studies have provided such mechanistic details for peptidoglycan hydrolyzing protein as such the Min et al., study is novel and of great interest. This work could, therefore, be potentially suited for Nature communication but while the biochemical work is well conducted the crystallographic data is of concern:

Major points:

Although the structures have been solved to high-resolution, several datasets display unusual

refinement statistics and notably:

-For the WT apo open form (6JMX) the validation report from the PDB indicates that numerous residues (near 70%) do not fit the electron density, the R/Rfree values are also slightly higher as compared to other structures at the same resolution. This could indicate a problem with data processing and probably a wrong space group choice notably because this dataset does not seem to be twinned. Indeed a reprocessing of data to space group P61 and a single refinement step (with Phenix.refine and using TLS) drops the Rfree value to 18% and improves greatly the RSRZ score.

-Another example is the structure of the WT apo closed form, solved at 1.6 Å resolution has very suspicious R and Rfree value (24 and 27%). This data seems slightly twinned but a wrong space group choice is also suspected. The electron density map is also of poor quality and very noisy (see in the solvent channel that is full of extra density). This might indicate also a wrong space group choice or another issue like ice ring but nothing is mentioned about it in this MS. A quick refinement step in the P321 space group (which might not be the correct one) yields indeed to still a high R/Rfree values but to a less noisy map. The authors should definitely reprocess and re-refined their structures with other space groups. Trying in P1 is also often useful in that case if the dataset is complete.

-Concerning the other datasets that probably were also processed in the wrong space group(s). The H216A apo and closed forms, as well as H247A apo-form, could probably be processed to a higher resolution (I/sigma of 5 and 7 in the last resolution shell) unless the detector was not moved close enough.

-in the same idea the Zn-SAD dataset cuts at 1.7 Å with an I/sigma of 7 in the last resolution shell is probably the best dataset in term of resolution (again if the detector was moved correctly) so why not building/refining the corresponding model.

Minor points:

-line 199 : "the crystallization data" should "the crystallographic data"

-line 278 : "verboden" can be indeed used in English but apart from German speaker most people will not understand it so please rephrase.

-line 381 : change "varint" to "variant"

Reviewer #1:

The manuscript reports several crystal structures of a new peptidoglycan hydrolase, Pgp3, from *Campylobacter jejuni*, and shows that the enzyme has DD-endopeptidase and DD-carboxypeptidase activity using synthetic peptidoglycan fragments as substrates. Pgp3 was identified by its sequence similarity to known members of the LytM type metallopeptidase family, for example the Csd peptidoglycan hydrolases (Csd1, Csd2, Csd4 and Csd6) from *Helicobacter pylori*, and Psp1 and Psp2 from *Campylobacter jejuni*. These enzymes are required for maintaining the helical cell shape in both pathogens. The current manuscript presents the crystal structure of Psp3 alone and of an inactive version with bound peptidoglycan fragments. The authors present convincing data on the importance of the catalytic Zn, which is expected based on the published structures of homologous hydrolases. They conclude that Pgp3 turns over peptidoglycan and that it is important to regulate the shape and virulence of *Campylobacter jejuni*. Whilst this is a solid structural work in my view the manuscript has major problems. In particular, they draw conclusions that are not supported by experimental data. There are many examples of proteins that don't have the same cellular function as an homolog, hence the function of a novel protein must be carefully determined and cannot be deduced from the function of its homologs.

Major concerns:

1. Title, Abstract and Concluding Remarks. The authors conclude that Pgp3 is "a cell shape determining protein" (line 340) and "an important enzyme that regulates the shape of ... *C. jejuni*." (line 349). This conclusion is not supported by experimental data. Hence, the authors should perform the following experiments:

1.1 Analyse the cell shape of a mutant strain that lacks the *pgp3* gene and assess if there are any cell shape defects.

→ We constructed a deletion mutant in *C. jejuni* and examined the morphology of the *pgp3* deletion mutant by TEM per Reviewer's comment. Indeed, the *pgp3* deletion mutant lost its helicity and displayed a slightly curved-rod morphology compared to wild-type (TEM images are given in Fig. 1c, the first and second rows).

1.2 Should the mutant strain have cell shape defects, complement the mutant strain by ectopic expression of the *pgp3* gene and test if the shape of wild-type cells is restored.

→ We attempted complementation by expressing the *pgp3* gene. The characteristic *C. jejuni* helical shape was restored (Fig. 1c, third row).

2. Concluding remarks (line 350): The authors conclude that Pgp3 is "an important enzyme that regulates the ... virulence of ... *C. jejuni*." This conclusion is not supported by experimental data. Hence, the authors should perform the following experiments:

1.1 Analyse the virulence of a mutant strain that lacks the *pgp3* gene, to test if there is any change in virulence compared to wild-type.

→ We performed an invasion assay using Caco-2 cells. The *pgp3* deletion mutant of *C. jejuni* exhibited significantly attenuated invasive ability compared to the wild-type (Fig. 1d).

1.2 If the mutant strain shows altered virulence, complement the mutant strain by ectopic expression of the *pgp3* gene and test if virulence is restored.

→ The invasive ability was restored after complementation by ectopic expression of the *pgp3* gene (Fig. 1d).

3. The authors also claim that "This enzyme turns over cell-wall peptidoglycan ...". However, they have shown activity only with soluble peptidoglycan fragments, and not with cell wall peptidoglycan. To test if Pgp3 is capable of digesting cell wall peptidoglycan they should:

3.1. test the activity against peptidoglycan from *Campylobacter*.

→ We had not previously prepared the sacculus of *C. jejuni*. We did that for the first time for this study. For reasons that are not obvious to us, the recovery of the sample was very poor and we could not assess the quality of the sacculus. The sacculus of *E. coli* is highly similar to that of *C. jejuni* with the difference of a minor *O*-acetylated component in that of *C. jejuni* (not seen in *E. coli*). We used a sacculus preparation from *E. coli* to document the _{D,D}-endopeptidase activity of Pgp3 (Supplementary Fig. 3). Increasing the concentration of Pgp3 diminishes the _{D,D}-crosslinks further (colored in blue, green and red in Supplementary Fig. 3). I hope the Reviewer finds this experiment acceptable.

3.2. test the activity against peptidoglycan in the presence of an MurNAc-L-Ala peptidoglycan amidase. This experiment is crucial to test the model presented in Supplemental Figure 7.

→ We are afraid that amidase treatment will significantly damage the peptidoglycan structure, so we wouldn't see the effect of amidase by the experiment. Because it is not clear whether amidase cleaves the peptidoglycan prior to Pgp3 as the reviewer mentioned, we deleted the figure (old Supplementary Fig. 7).

3.3 compare the peptidoglycan composition of a *pgp3* mutant strain with that of wild-type.

→ As we explained under 3.1, we were not able to prepare the intact sacculus of *C. jejuni*, hence this experiment could not be attempted. Nonetheless, we documented the enzymatic activity with the *E. coli* sacculus.

Minor point:

4. I would not use the term "bi-functional enzyme" (line 343) to characterize an enzyme with _{DD}-endopeptidase and _{DD}-carboxypeptidase activity, because both activities involve the same active site and similar substrates. A bifunctional enzyme would rather be one that has two different active sites/domains catalyzing different reactions.

→ We removed the "bi-functional enzyme" term as suggested. The word "bifunctional" used in other sentences (lines 37 and 109) were also rephrased.

5. line 342. The synthetic peptidoglycan fragments used were not perfectly "authentic", as they were methylated at the MurNAc residue, which would not occur in natural peptidoglycan.

→ In the natural polymeric peptidoglycan, one would have the structure of the extended polysaccharide in place of the C1-methoxy. This said, methoxy (as opposed to a hydroxy) was

introduced to C1 to fix the anomeric configuration to beta, which is found in the natural peptidoglycan. The word “authentic” was used in its meaning that the quality/accuracy of the chemical structure was established by synthesis and by the subsequent detailed spectroscopic characterization of the synthetic sample. As it is just easier to address the comment of the Reviewer by removing the descriptor “authentic”, this is exactly what we did on revision. We trust that this is acceptable.

6. line 203 and elsewhere. It should be added that the Ala1 residue has L-configuration.
→ Done as suggested (line 232).

Reviewer #2:

This manuscript describes an interesting, important and topical area of science; the maintenance of bacterial viability and identification of potential targets for new antibiotics.

The study is truly multidisciplinary with structural biology and molecular simulations playing key roles.

I am only able to comment on the simulations as an expert; regarding the X-ray structures I can comment as a knowledgeable inexpert!

I think overall the study is of high quality and the data are interesting.

I think the simulation models and methods are appropriate. However I am unclear as to how many independent simulations of each system were conducted. I can only assume that this is more than 2, if not then it should be to demonstrate reproducibility. I have no other real concerns.

→ We performed a number of simulations in each case to explore reproducibility, and indeed they were reproducible. What we gave in the manuscript for each system was the final “production” simulation (for 250 ns) .

Minor comments.

- Line 173 'Both tightly bonds to the active site' - please rephrase this to make it clear you are referring to the shape fit of the molecules here and not energetics.

→ We have rephrased the statement “*Both molecules bound to the active site of Pgp3 via several hydrogen bonds and salt-bridge interactions between the carboxyl groups of tartrate or citrate and the Zn²⁺ ion (Figs. 2A and 2B).*” (lines 200-202).

- Line 188 ' pulling the β7-β8-β9 sheet by 49 ° - a distance measurement might be useful here too?

→ The distance has been described in terms of RMSD here “*However, both structures showed a large deviation (RMSD 6.98 Å) in two regions (residues Glu99-Phe109 and Thr159-Val164) near the active-site.*” As requested by the reviewer, we have also included the distance between center of masses of the β7-β8-β9 sheet in the open and closed conformation. In place of this line “*Interestingly, Arg158 showed large conformational change, pulling the β7↑-β8↓-β9↑ sheet by*

49° in ...”, it has been changed to “Interestingly, Arg158 showed large conformational change, pulling center of mass of $\beta 7 \uparrow$ - $\beta 8 \downarrow$ - $\beta 9 \uparrow$ sheet by 49° (and 6 Å) in ...” (lines 214-215).

- 252-253 'indicating that Wat-1 is the hydrolytic water' please rephrase this - it only suggests that it is likely to be the key water, it is not conclusive, please make this clear.

→ At the suggestion of the reviewer, this statement has now been rephrased as “*However, the crystal structure of the H247A variant in complex with the substrate clearly showed that Wat-2 was replaced by the incoming substrates, concluding that Wat-1 is the hydrolytic water that attacks the scissile carbonyl group of substrates.*” (lines 280-282).

Supporting information

- 'Protein expression and purification' <-- correct typographical error

→ Corrected as indicated (line 32).

- Line 76 (gold standard of water models) <-- please remove this as accuracy of water models is very much dependent upon the quantities being measured.

→ The statement “(gold standard of water models)” was removed on revision.

For the REMD simulations please explain why the lowest temperature was chosen to be 270 K when the other, traditional MD simulations are performed at 300 K.

→ The selection of the temperature range for REMD itself makes an interesting study as has been documented by several groups. The most common references are related to protein folding and conformational change studies (Sugita & Okamoto, *Chem. Phys. Lett.*, 1999, 314(2), pp. 141-151). While it is standard practice to use the temperature range between 260–700 K for protein-folding experiments, there is only limited study on impact of temperature range on enhanced sampling for protein-ligand complexes. We selected a wide range of temperature to ensure enhanced sampling as well as correlation between lower temperature experimental information with simulation results, like the study by Ravindranathan et al. *J. Am. Chem. Soc.* 2006, 128(17), pp. 5786-5791.

Reviewer #3:

Min et al., report several high-resolution X-ray structures and the biochemical characterization of the Pgp3 peptidase from *C. jejuni*. This bacterium is a human pathogen with a high incidence in term of associated guts disease. The reason for the atypical helical shape of this bacterium is not completely understood. Identifying key player(s) responsible for this shape and understanding their role could potentially pave the way to identify new drug target.

In this study, the authors solved numerous crystal structure of the Pgp3 protein in its apo and closed forms and in the presence of peptidoglycan synthetic fragments. Thanks to these structures a very interesting mechanism of peptidoglycan recognition and hydrolysis have been proposed. Rare studies have provided such mechanistic details for peptidoglycan hydrolyzing

protein as such the Min et al., study is novel and of great interest. This work could, therefore, be potentially suited for Nature communication but while the biochemical work is well conducted the crystallographic data is of concern:

Major points:

Although the structures have been solved to high-resolution, several datasets display unusual refinement statistics and notably:

-For the WT apo open form (6JMX) the validation report from the PDB indicates that numerous residues (near 70%) do not fit the electron density, the R/R_{free} values are also slightly higher as compared to other structures at the same resolution. This could indicate a problem with data processing and probably a wrong space group choice notably because this dataset does not seem to be twinned. Indeed a reprocessing of data to space group to *P6₁* and a single refinement step (with Phenix.refine and using TLS) drops the R_{free} value to 18% and improves greatly the RSRZ score.

→ Thanks for great comments. When we checked all data set by *P1* processing and examined all possible space groups, we found that the reviewer is correct. As suggested by the reviewer, the five *P3₁* dataset (WT apo open form, H216A apo open form, H247A apo form, H247A tetra-tri peptide bound form, and H247A penta peptide bound form) were changed to *P6₁*. After refinement with *P6₁* with TLS, R/R_{free} values and RSRZ score were significantly improved; WT apo open form R/R_{free} values (21.7/23.4% to 17.2/19.9%) and RSRZ score (67.6 to 6.2); H216A apo open form R/R_{free} values (19.1/23.4% to 18.2/21.3%) and RSRZ score (18.7 to 6.5); H247A apo open form R/R_{free} values (20.7/23.5% to 17.1/19.7%) and RSRZ score (33.5 to 7.5); H247A tetra-tri peptide bound form R/R_{free} values (20.8/24.3% to 16.1/19.9%) and RSRZ score (5.7 to 8.6); H247A penta peptide bound form R/R_{free} values (19.9/25.1% to 17.8 / 22.6%) and RSRZ score (7.1 to 8.7). The protein structures were highly similar to previous structures after the re-refinement and all of data from the revised refinement is updated in Supplementary Table 1.

-Another example is the structure of the WT apo closed form, solved at 1.6 Å resolution has very suspicious R and R_{free} value (24 and 27%). This data seems slightly twinned but a wrong space group choice is also suspected. The electron density map is also of poor quality and very noisy (see in the solvent channel that is full of extra density). This might indicate also a wrong space group choice or another issue like ice ring but nothing is mentioned about it in this MS. A quick refinement step in the *P321* space group (which might not be the correct one) yields indeed to still a high R/R_{free} values but to a less noisy map. The authors should definitely reprocess and re-refined their structures with other space groups. Trying in *P1* is also often useful in that case if the dataset is complete.

→ Reviewer is correct. As a result, the *P3₂* space group was changed to *P3₂21*. The R/R_{free} values were also improved from 24.5/27.9% to 17.3/20.6%. All of data from the refinement is updated in Table S1.

-Concerning the other datasets that probably were also processed in the wrong space group(s). The H216A apo and closed forms, as well as H247A apo-form, could probably be processed to a

higher resolution (I/sigma of 5 and 7 in the last resolution shell) unless the detector was not moved close enough.

→ We realized that the detector distance was not enough to collect the high-resolution data for the three dataset (H216A apo form, H216A closed form, and H247A apo-form). Thus, we could not refine the structures to higher resolution.

-in the same idea the Zn-SAD dataset cuts at 1.7 Å with an I/sigma of 7 in the last resolution shell is probably the best dataset in term of resolution (again if the detector was moved correctly) so why not building/refining the corresponding model.

→ We tried to refine the Zn-SAD dataset. It turned out that the Zn-SAD dataset corresponds to WT apo closed form. However, we could not get the model with better R/R_{free} values because of poor data quality at highest resolution shell. However, we refined the data and additionally deposited the model (R/R_{free} = 21.7 / 26.0).

Minor points:

-line 199 : “the crystallization data” should “the crystallographic data”

→ Corrected as indicated (line 227).

-line 278 : “verboten” can be indeed used in English but apart from German speaker most people will not understand it so please rephrase.

→ The word “verboten” was replaced with “prohibited” (line 307).

-line 381 : change “varint” to “variant”

→ Corrected as indicated (line 410).

REVIEWERS' COMMENTS:

Reviewer #1 (Remarks to the Author):

The manuscript has significantly improved by adding data showing that the deletion of the *pgp3* gene impairs helical cell morphology and invasion into Caco-2 cells, and the complementation of these phenotypes by ectopic expression of *pgp3*. Unfortunately, they did not include peptidoglycan composition of the *pgp3* mutant, but they added biochemical data on the activity of Pgp3 against *E. coli* sacculi. Hence, I am satisfied with the conclusions presented in the revised manuscript.

Reviewer #3 (Remarks to the Author):

The first version of the Min et al., manuscript describing the characterization of the peptidoglycan peptidase 3 from *C. jejuni* had serious problems in the structural data processing. The authors have now corrected all the issues concerning the crystallographic data. It is worth mentioning that the authors also did excellent work concerning the in vivo experiments and the generation of the *C. jejuni* mutant. I, therefore, recommend the publication of this manuscript.

with my best regards

M.Blaise